# A web server for comparative analysis of single-cell RNA-seq data

Amir Alavi [1], Matthew Ruffalo [1], Aiyappa Parvangada[1], Zhilin Huang[1] & Ziv Bar-Joseph[1,2]

Single cell RNA-Seq (scRNA-seq) studies profile thousands of cells in heterogeneous environments. Current methods for characterizing cells perform unsupervised analysis followed by assignment using a small set of known marker genes. Such approaches are limited to a few, well characterized cell types. We developed an automated pipeline to download, process, and annotate publicly available scRNA-seq datasets to enable large scale supervised characterization. We extend supervised neural networks to obtain efficient and accurate representations for scRNA-seq data. We apply our pipeline to analyze data from over 500 different studies with over 300 unique cell types and show that supervised methods outperform unsupervised methods for cell type identification. A case study highlights the usefulness of these methods for comparing cell type distributions in healthy and diseased mice. Finally, we present scQuery, a web server which uses our neural networks and fast matching methods to determine cell types, key genes, and more.

[1] Computational Biology Department, School of Computer Science, Carnegie Mellon University, Pittsburgh, PA 15213, USA. [2] Machine Learning Department, School of Computer Science, Carnegie Mellon University, Pittsburgh, PA 15213, USA. These authors contributed equally: Amir Alavi, Matthew Ruffalo. Correspondence and requests for materials should be addressed to Z.B-J. (email: zivbj@cs.cmu.edu)

Single-cell RNA sequencing (scRNA-seq) has recently emerged as a major advancement in the field of transcriptomics[1]. Compared to bulk (many cells at a time) RNA-seq, scRNA-seq can achieve a higher degree of resolution, revealing many properties of subpopulations in heterogeneous groups of cells[2]. Several different cell types have now been profiled using scRNA-seq leading to the characterization of subtypes, identification of new marker genes, and analysis of cell fate and development[3–5].

While most work attempted to characterize expression profiles for specific (known) cell types, more recent work has attempted to use this technology to compare differences between different states (for example, disease vs. healthy cell distributions) or time (for example, sets of cells in different developmental time points or age)[6,7]. For such studies, the main focus is on the characterization of the different cell types within each population being compared, and the analysis of the differences in such types. To date, such work primarily relied on known markers[8] or unsupervised (dimensionality reduction or clustering) methods[9]. Markers, while useful, are limited and are not available for several cell types. Unsupervised methods are useful to overcome this, and may allow users to observe large differences in expression profiles, but as we and others have shown, they are harder to interpret and often less accurate than supervised methods[10].

To address these problems, we have developed a framework that combines the idea of markers for cell types with the scale obtained from global analysis of all available scRNA-seq data. We developed scQuery, a web server that utilizes scRNA-seq data collected from over 500 different experiments for the analysis of new scRNA-Seq data. The web server provides users with information about the cell type predicted for each cell, overall cell-type distribution, set of differentially expressed (DE) genes identified for cells, prior data that is closest to the new data, and more.

Here, we test scQuery in several cross-validation experiments. We also perform a case study in which we analyze close to 2000 cells from a neurodegeneration study[6], and demonstrate that our pipeline and web server enable coherent comparative analysis of scRNA-seq datasets. As we show, in all cases we observe good performance of the methods we use and of the overall web server for the analysis of new scRNA-seq data.

## Results

**Pipeline and web server overview**. We developed a pipeline (Fig. 1) for querying, downloading, aligning, and quantifying scRNA-seq data. Following queries to the major repositories (Methods), we uniformly processed all datasets so that each was represented by the same set of genes and underwent the same normalization procedure (RPKM). We next attempt to assign each cell to a common ontology term using text analysis (Methods and Supporting Methods). This uniform processing allowed us to generate a combined dataset that represented expression experiments from more than 500 different scRNA-seq studies, representing 300 unique cell types, and totaling almost 150 K expression profiles that passed our stringent filtering criteria for both expression quality and ontology assignment (Methods). We next used supervised neural network (NN) models to learn reduced dimension representations for each of the input profiles. We tested several different types of NNs including architectures that utilize prior biological knowledge[10] to reduce overfitting as well as architectures that directly learn a discriminatory reduced dimension profile (siamese[11] and triplet[12] architectures). Reduced dimension profiles for all data were then stored on a web server that allows users to perform queries to compare new scRNA-seq experiments to all data collected so far

to determine cell types, identify similar experiments, and focus on key genes.

**Statistics for data processing and downloads**. To retrieve all available scRNA-seq data, we queried the two largest databases, GEO and ArrayExpress, for scRNA-seq data. Supplementary Figure 1 presents screenshots of queries to the NCBI GEO and ArrayExpress databases similar to the ones we used here, though our queries utilized automated APIs instead of the web interfaces shown in these figures. Supplementary Figure 2 and Fig. 2, respectively, show study and cell counts by month, with respect to the "release date" data provided by GEO and ArrayExpress. As can be seen, while cell counts increase over time, there is a lag in availability of raw data and author-processed supplementary data available through NCBI GEO and ArrayExpress systems. Since our pipeline is automated, we expect to be able to collect and analyze much more data over the next several months.

Our "mouse single-cell RNA-seq" query matched a total of 193,414 cells, of which 151,084 have raw data and 29,216 had only author-processed data. We used established ontologies to determine the cell type that was profiled (Methods) for each cell expression dataset we downloaded. Of the 2481 unique descriptors we obtained for all cells, 1909 map to at least one term in the cell ontology. Of the 5010 distinct cell ontology terms (restricted to the CL and UBERON namespaces), 331 are assigned at least one cell expression profile.

Of the 151,084 cells for which raw data are available, 114,249 had alignment rates above our cutoff of 40%. Of these 114,249, we identified 2473 raw data files that contained reads from multiple cells, but lacked any metadata that allowed us to assign reads to individual cells. This leaves 143,465 cells that are usable for building our scRNA-seq database.

**Neural networks for supervised dimensionality reduction**. We trained several different types of supervised NNs. These included models with the label matching a cell type as the output (with the layer before last serving as the reduced dimension)[10] and models that directly optimize a discriminatory reduced dimension layer (using as input pairs or triplets of matched and unmatched profiles). See Supplementary Figs. 10, 12, and 13 for details. Some of the models utilized prior biological knowledge as part of the architecture to reduce overfitting (including protein–protein and protein–DNA interaction data, termed PPI and TF respectively, and hierarchical GO assignments) while others did not (dense). We experimented with various hyperparameters (Methods), and all of our neural network models were trained for up to 100 epochs, with training terminated early, if the model converged. All models converged sooner than the full 100 epochs (Supplementary Fig. 4). Performance on a held out validation set was assessed after each epoch during training. The final weights chosen for each model were those at the end of the epoch with the lowest validation loss out of the 100 epochs. For triplet networks, we also monitored the fraction of "active triplets" in each batch as a selection criterion[33]. Most models trained in minutes (Supplementary Table 1).

Prior work[10] has shown that NN weights can be used to identify relevant groupings of genes. Here we focused on the accuracy of the learned networks. After training each of our neural embedding models, we evaluated their performance using retrieval testing as described in Methods. The training set we used contained 36,473 cells, while the test ("query") set consisted of 2,330 cells. Cells used for testing are completely disjoint from the set of cells that were used for training and come from different studies so that batch effects and other experimental artifacts do not affect performance and evaluation. The results of this retrieval

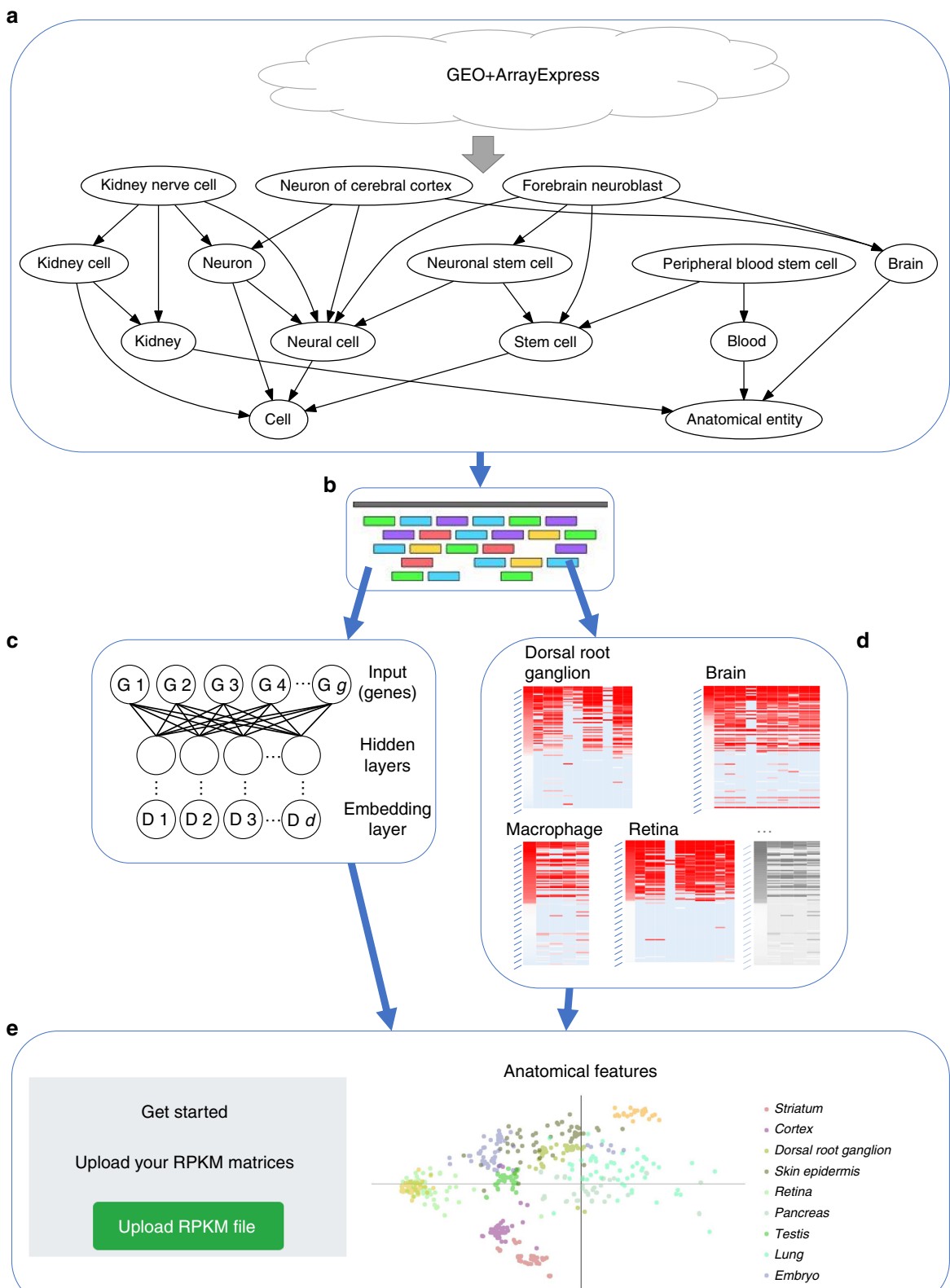

**Fig. 1** Pipeline for large-scale, automated analysis of scRNA-seq data. **a** Bi-weekly querying of GEO and ArrayExpress to download the latest data, followed by automatic label inference by mapping to the Cell Ontology. **b** Uniform alignment of all datasets using HISAT2, followed by quantification to obtain RPKM values. **c** Supervised dimensionality reduction using our neural embedding models. **d** Identification of cell-type-specific gene lists using differential expression analysis. **e** Integration of data and methods into a publicly available web application

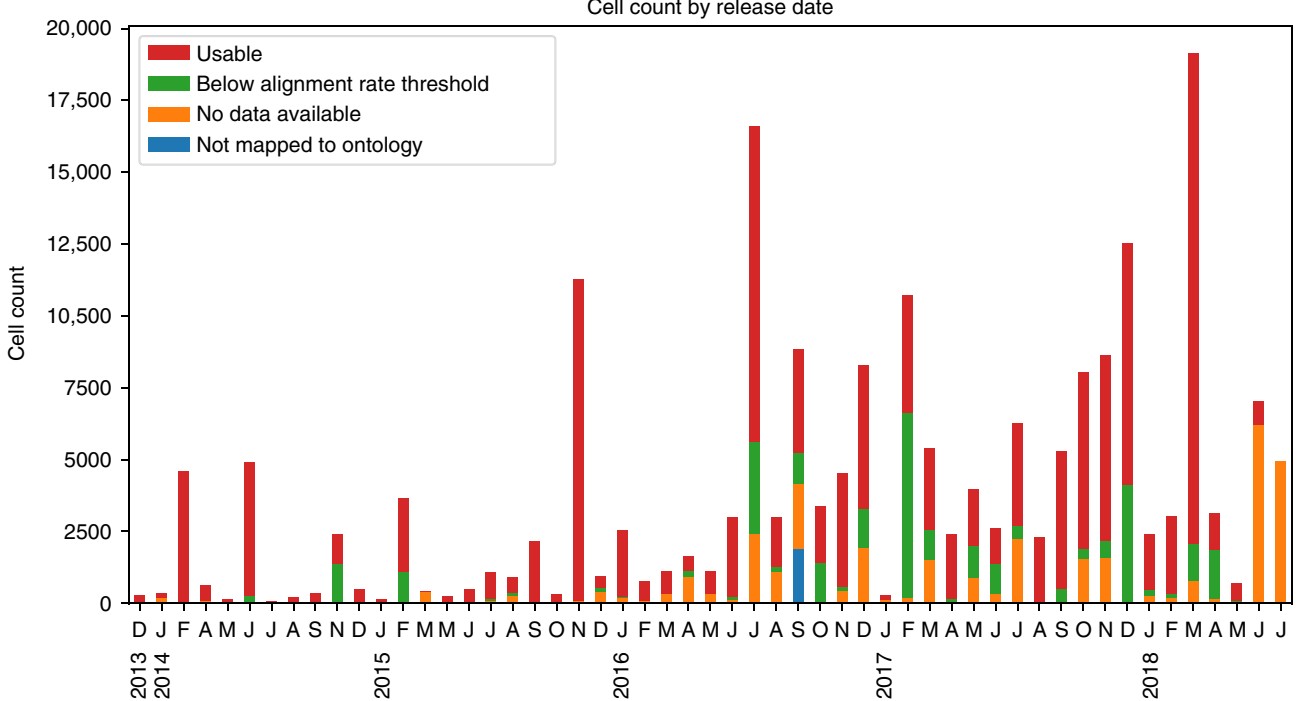

**Fig. 2** Monthly cell count available on GEO and ArrayExpress. Cell counts by month, separated into four categories: usable, below our alignment rate threshold, no raw or author-processed data available, and unmapped to ontology terms

| Architecture type | Model | Hematop-oietic stem cell | Osteocyte | Neuron | Neural | Embryo | Retina | Skin epidermis | Weighted average (higly rep. cell types) |
|---|---|---|---|---|---|---|---|---|---|
| | # in query | 91 | 45 | 162 | 81 | 45 | 250 | 678 | |
| | # in database | 734 | 481 | 421 | 523 | 250 | 340 | 1097 | |
| Cell type classifier | PT dense 1136 100 | 0.683 | **1.000** | 0.931 | 0.574 | 0.145 | 0.909 | **0.621** | **0.623** |
| | PT dense 1136 500 100 | **0.798** | 0.977 | 0.905 | 0.930 | 0.091 | 0.909 | 0.573 | 0.586 |
| Triplet | PT (frac active) ppitf 1136 500 100 | 0.382 | 0.951 | 0.955 | 0.644 | 0.493 | 0.941 | 0.598 | 0.594 |
| | PT (val loss) ppitf 1136 500 100 | 0.308 | 0.961 | **0.963** | 0.516 | **0.640** | **0.942** | 0.552 | 0.550 |
| Siamese | PT dense 1136 500 100 | 0.510 | 0.981 | 0.937 | **0.996** | 0.258 | 0.896 | 0.426 | 0.483 |
| | PT ppitf 1136 100 | 0.113 | 0.767 | 0.385 | 0.952 | 0.082 | 0.870 | 0.159 | 0.224 |
| N/A | PCA 100 | 0.696 | 0.946 | 0.863 | 0.889 | 0.167 | 0.901 | 0.488 | 0.494 |
| | Original data | 0.019 | 0.976 | 0.539 | 0.823 | 0.146 | 0.797 | 0.062 | 0.109 |

**Fig. 3** Neural embedding retrieval testing results. Retrieval testing results of various architectures, as well as PCA and the original (unreduced) expression data. Scores are MAFP (mean average flexible precision) values (Supporting Methods). "PT" indicates that the model had been pretrained using the unsupervised strategy (Supporting Methods). "Ppitf" refers to architectures based on protein–protien and protein–DNA interactions (Supporting Methods, Supplementary Fig. 12). Numbers after the model name indicate the hidden layer sizes. For example, "dense 1136 500 100" is an architecture with three hidden layers. The metrics in parenthesis for the triplet architectures indicate the metric used to select the best weights over the training epochs. For example, "frac active" indicates that the weights chosen for that model were the ones that had the lowest fraction of active triplets in each mini-batch. We highlight the best performing model in each cell type with a bolded value. We can see that in every column, the best model is always one of our neural embedding models. The final column shows the weighted average score over those cell types, where the weights are the number of such cells in the query set. The best neural embedding model (PT dense 1136 100, top row) outperformed PCA 100 (0.623 vs. 0.494) with a $p$-value of $1.253 \times 10^{-41}$ based on two-tailed $t$-test. Source data are provided as a Source Data file

testing for a selection of architecture types and cell types are shown in Fig. 3. These models have been trained using the data that we processed ourselves in addition to data from studies that only had author-processed data available (these required missing-data imputation, see Methods) though similar results were obtained on models trained using only our own processed data (Supplementary Fig. 6). It is common to assess retrieval performance with mean average precision (MAP). Here, each value in the table represents the mean average flexible precision

(MAFP, Supporting Methods), which allows for scores between 0 and 1 for matching a cell type to a similar or parent type (for example, a cortex cell matched to brain, Methods).

The best scoring model achieved a weighted average (accross all query cell types) MAFP of 0.576, which is very high when considering the fact that this was a 45-way classification problem (while our database contains over 300 cell types, only 45 types had independent data from multiple studies and these were used for the analysis discussed here). When restricting the analysis to

the six cell types for which we had more than 1000 cells in our database, results for this model further improved to 0.623, which is significantly better than PCA 100 (paired $t$-test $p$-value: $1.253 \times 10^{-41}$). A paired $t$-test comparison on the full query set of cells is shown in Supplementary Table 2. This top performing model (first row in Fig. 3) employed a dense architecture (two-hidden-layer perceptron network). The next best model overall (based on the weighted average) was a similarly defined and performing dense architecture with three hidden layers (in Source Data), followed by a PPITF architecture with three hidden layers (the first one being sparsely connected to the input based on prior biological knowledge) trained as a triplet network (third row in Fig. 3). We also see that for specific cell types, other neural networks perform better. Specifically, triplet networks perform best for neuron, embryo, and retina. Siamese architectures perform the best in the neural cell type. We also note that the best performing models were those that were pretrained with an unsupervised strategy (a full table with result from over 100 models, including those from models without pretraining are available on our web server). Finally, as is clear from the last two rows, supervised neural network embeddings consistently outperform PCA and the original data in the retrieval task. Even when we consider only those cell types that are rare in our training data (<1% of total training population), we again see our neural embedding models outperforming PCA and original data (Supplementary Fig. 5).

**Functional analysis of cell-type-specific DE genes**. We further used our ontology assignments to identify cell-type-specific genes (Methods). The differential expression analysis we conducted is based on multiple studies for each cell type. Our procedure performs DE analysis for each study, for each cell type independently and then combines the results. This method ensures that resulting DE genes are not batch or lab related but rather real DE for the specific cell type. We used this method to identify cell-type-specific genes for 66 cell types. The number of significant (<0.05 FDR adjusted $p$-value) DE genes for each cell type ranged from 31 for embryonic stem cells to 7576 for B cells. The full list of DE genes can be found on the supporting web server.

To determine the accuracy of the DE genes and to showcase the effectiveness of the automated processing and ontology assignments, we performed Gene Ontology (GO) enrichment analysis on the set of DE genes for each cell type (Supporting Methods). Results for a number of the cell types are presented in Table 1. As can be seen, even though each of the cell-type data we used combined multiple studies from different labs, the categories identified for all of the cell types are highly specific indicating that the automated cell-type assignment and processing were able to correctly group related experiments.

As an example, the top three enriched terms for "retina" are "nervous system process" ($p = 7.63 \times 10^{-9}$), "sensory perception of light stimulus" ($p = 7.63 \times 10^{-9}$), and "visual perception" ($p = 7.63 \times 10^{-9}$). Cells of dorsal root ganglion are sensory neurons as reflected in Table 1 with terms such as "sensory perception of pain" ($p = 3.54 \times 10^{-6}$) and "detection of temperature stimulus" ($p = 9.86 \times 10^{-6}$). For "T cell," nine of the top ten terms are related to immune response and specific aspects of the T cell-mediated immune system. Complete results are available from the supporting web server.

**Mouse brain case study**. To test the application of our pipeline we used it to study a recent scRNA-seq neurodegeneration dataset that was not included in our database[6]. This study profiled 2208 microglial cells extracted from the hippocampus of the CK-p25 mouse model of severe neurodegeneration. In the CK-p25 mouse model, the expression of p25, a calpain cleaved kinase activator, is induced and results in Alzheimer's disease-like pathology. In the original study, the microglial cells were extracted from control and CK-p25 mice from four time points: before p25 induction (3 months old), and 1, 2, and 6 weeks after induction (3 months 1 week, 3 months 2 weeks old, and 4 months 2 weeks old, respectively). The goal of the study was to compare the response of microglial cells to determine distinct molecular sub-types, uncover disease-stage-specific states, and further characterize the heterogeneity in microglial response. We used the raw read data to perform alignment and quantification (Methods) resulting in 1990 cells that passed our alignment thresholds and were used for the analysis that follows.

We used these data to test several aspects of the method, pipeline, and website. We performed a complete analysis of the roughly 2000 cells using the scQuery web server (see below), which only took a few minutes. We first compared the supervised (using NN) and unsupervised dimensionality reduction. The cells were transformed to a lower dimensional space using the "PT dense 1136 100" NN followed by t-SNE to get them to 2 dimensions. We compared this to a completely unsupervised dimensionality reduction, as was done in the original paper. Supplementary Fig. 7 presents the results of this analysis. We observe that the supervised method is able to better account for the differences between the two populations of healthy and disease cells.

**Cell-type classifications**. We next performed retrieval analysis by using the mouse brain cells as queries against our large database of labeled cells. We classified each query cell based on the most common label in its 100 nearest neighbors in the database (Methods). The results of this cell-type classification can be seen in Fig. 4.

We next compared the cell assignments for three different groups. An early time point (3 months old) in which the healthy and disease mouse models are not expected to diverge, a later time point (4 months 2 weeks old) in which differences are expected to be pronounced, and all data collected from the healthy and disease data. As can be seen in Fig. 4a, our assignment indeed reflects these stages with a much more significant difference for the later time point compared to the earlier one, with the entire dataset (which includes more intermediate points) in the middle. Focusing on the later time point, Fig. 4b shows the cell-type distribution. Several of the cell types identified by the method correspond to brain cells (brain, cortex, meningeal cluster) while others are related to blood and immune response (bone marrow, macrophage). The most common classification among the query cells was "fibroblast." Recent studies have shown that fibroblast-like cells are common in the brain[13], and that brain fibroblast cells can express neuronal markers[14].

As can be seen, the main difference observed between the disease and healthy mice is the increase in the immune system-related types of "bone marrow" and "macrophage" cells in the disease model. We believe that while the method labeled these cells as macrophages, they are actually microglia cells that were indeed the cells the authors tried to isolate. To confirm this, we analyzed sets of marker genes that are distinct for macrophages and for microglia[15]. Supplementary Fig. 9 shows that indeed, for the cells identified by the method the expressed markers are primarily microglia markers.

The main reason that the method identified them as macrophages is the lack of training data for microglial cells in our database (our train data of high-confidence cell types contains no microglial cells and 273 macrophage cells; our full

**Table 1 Results of GO enrichment analysis**

| Cell type | Experiments | GO ID | Name | *p*-Value |
|---|---|---|---|---|
| Retina | 2 | GO:0050877 | Nervous system process | 7.63e−09 |
| | | GO:0050953 | Sensory perception of light stimulus | 7.63e−09 |
| | | GO:0007601 | Visual perception | 7.63e−09 |
| | | GO:0007423 | Sensory organ development | 8.69e−09 |
| | | GO:0060041 | Retina development in camera-type eye | 8.69e−09 |
| | | GO:0007600 | Sensory perception | 4.09e−08 |
| | | GO:0001654 | Eye development | 3.96e−07 |
| | | GO:0003008 | System process | 6.67e−07 |
| | | GO:0009584 | Detection of visible light | 6.86e−07 |
| | | GO:0043010 | Camera-type eye development | 1.23e−06 |
| Dorsal root ganglion | 3 | GO:0019233 | Sensory perception of pain | 3.54e−06 |
| | | GO:0016048 | Detection of temperature stimulus | 9.86e−06 |
| | | GO:0031175 | Neuron projection development | 2.41e−04 |
| | | GO:0050965 | Detection of temperature stimulus involved in... | 2.51e−04 |
| | | GO:0050877 | Nervous system process | 2.51e−04 |
| | | GO:0050961 | Detection of temperature stimulus involved in... | 2.51e−04 |
| | | GO:0048666 | Neuron development | 3.49e−04 |
| | | GO:0009581 | Detection of external stimulus | 3.79e−04 |
| | | GO:0009582 | Detection of abiotic stimulus | 3.79e−04 |
| | | GO:0007600 | Sensory perception | 4.31e−04 |
| Skin epidermis | 2 | GO:0009888 | Tissue development | 2.99e−06 |
| | | GO:0043588 | Skin development | 2.99e−06 |
| | | GO:0042303 | Molting cycle | 9.57e−05 |
| | | GO:0042633 | Hair cycle | 9.57e−05 |
| | | GO:0030177 | Positive regulation of Wnt signaling pathway | 1.16e−04 |
| | | GO:0042476 | Odontogenesis | 1.16e−04 |
| | | GO:0009653 | Anatomical structure morphogenesis | 1.87e−04 |
| | | GO:0022404 | Molting cycle process | 2.37e−04 |
| | | GO:0030111 | Regulation of Wnt signaling pathway | 2.37e−04 |
| | | GO:0022405 | Hair cycle process | 2.37e−04 |
| T cell | 4 | GO:0002376 | Immune system process | 4.68e−10 |
| | | GO:0006955 | Immune response | 3.86e−08 |
| | | GO:0046649 | Lymphocyte activation | 1.25e−07 |
| | | GO:0045321 | Leukocyte activation | 1.25e−07 |
| | | GO:0002682 | Regulation of immune system process | 1.25e−07 |
| | | GO:0042110 | T cell activation | 2.75e−07 |
| | | GO:0050776 | Regulation of immune response | 2.84e−07 |
| | | GO:0002684 | Positive regulation of immune system process | 3.82e−07 |
| | | GO:0001775 | Cell activation | 4.54e−07 |
| | | GO:0002252 | Immune effector process | 6.38e−07 |

Top 10 enriched "GO: Biological Process" terms using top 50 upregulated DE genes for each cell type (FDR adjusted *p*-values). "Experiments" refers to the number of studies used to compute differentially expressed genes (Methods)

database contains only 44 microglial cells compared to 603 macrophage cells). Still, the result that disease samples contain more immune cells, which is only based on our analysis of scRNA-seq data (without using any known immune markers) indicates that as more scRNA-seq studies are performed and entered into our database, the accuracy of the results would increase.

**Comparison to macrophage differentially expressed genes**. We further characterized the gene expression within these microglial cells by comparing the gene expression in the query cells to our cell-type-specific DE genes (Methods). We uploaded the RPKM gene expression values for 256 microglial cells from late-stage neurodegenerative mice (6 weeks after p25 induction) to our web server[6]. Given the hits from the retrieval database, we then selected a cluster of these that were enriched for "macrophage", and then viewed the expression (logRPKM) of our previously calculated cell-type-specific marker genes as a heatmap for a subset of these cells. A cropped screenshot of the interactive heatmap provided by the web server is shown in Supplementary

Fig. 8, where we see that there is a rough agreement in upregulation and downregulation trends of the set of macrophage-specific DE genes we identified between the cells in the database (leftmost column) and the query (microglial) cells. We would expect to see this expression pattern for these genes in these cells, as microglia are distinct from macrophages, but are related in function.

Among the upregulated genes selected for "macrophage", we see genes that are also found to be upregulated in late-response clusters including a microglial marker (*Csf1r*), a gene belonging to the chemokine superfamily of proteins (*Ccl4*), a major histo-compatability complex (MHC) class II gene (*Cd74*), and other genes related to immune response (*Lilrb4a, Lgals3*)[6], (magenta highlights in Supplementary Fig. 8). We also see that a different microglial marker (*C1qa*)[16] is upregulated. In addition to re-identifying genes of interest from the original study, our method is also able to highlight additional genes that are biologically relevant. These are highlighted in yellow in Supplementary Fig. 8 and include more chemokines (*Ccl2, Ccl6, Ccl9*), another MHC class II gene (*H2-ab1*), and other cell surface antigens (*Cd14, Cd48, Cd52, Cd53*).

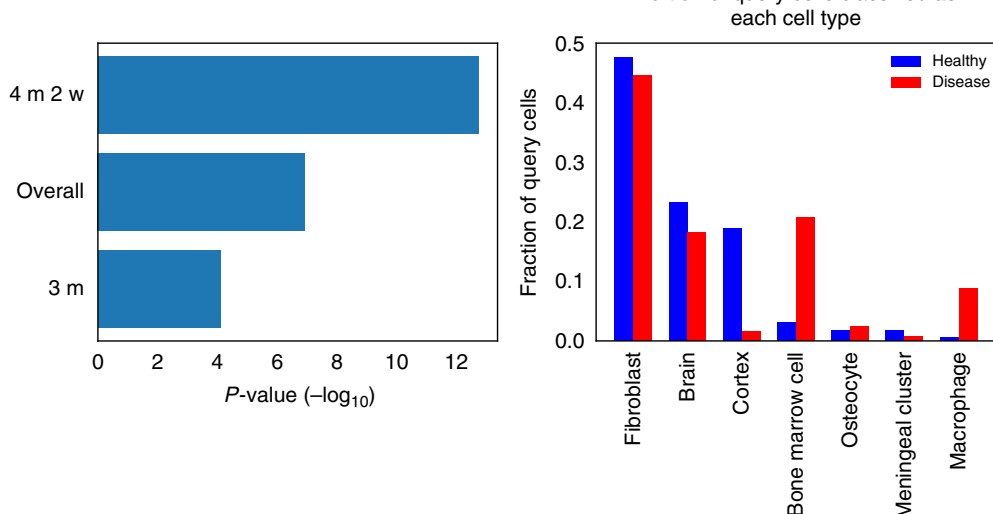

**Fig. 4** Analysis of mouse neurodegeneration dataset, late-response cells. **a** p-Values of the difference in cell-type classification distributions (healthy vs. disease cells) for different time points. Three months was the initial time point in the study, and 4 months 2 weeks was the last time point. "Overall" is the pool of all 1990 cells. The p-values are from conducting Fisher's exact test (for "overall", the p-value was simulated based on $1 \times 10^7$ replicates). **b** Classification distribution for late-stage cells (4 months 2 weeks), showing an increase in immune-related cell types in the disease cell population

**Query and retrieval**. To enable users to compare new scRNA-seq data to the data we processed, and to determine the composition of cell types in such samples, we developed a web application. After processing their data (Methods) users can upload it to the server. Next, uploaded data are compared to all studies stored in the database. For this, we use approximate nearest neighbor approaches to match these to the data we have pre-processed. Embedding in the reduced dimension representation and the fast matching queries of thousands of cells against hundreds of thousands of database cells can be performed in minutes.

Figure 5 presents an example of a partial analysis of newly uploaded data by the web server. The web server clusters cells based on their matched types (Fig. 5a), plots their 2D embedding with respect to all other cell types in the database (Fig. 5b), highlights the top represented ontology terms in the uploaded cells (Fig. 5c), and provides additional information about specific studies that it matched to the uploaded data (Fig. 5d).

**Discussion**

We developed a computational pipeline to process all scRNA-seq data deposited in public repositories. We have identified over 500 studies of scRNA-seq data. For each, we attempted to download the raw data and to assign each cell to a restricted ontology of cell types. For cells for which this information existed, we uniformly processed all reads, ran them through a supervised dimensionality reduction method based on NNs, and created a database of cell-type profiles. Using the scQuery server, users can upload new data, process it in the same way, and then compare it to all collected scRNA-seq data. We view the main goal of scQuery as a way to assist experimentalists who are performing exploratory analysis of new scRNA-Seq studies or re-analyzing existing studies. scQuery addresses several issues in the analysis of new scRNA-Seq data by automating the annotation process based on prior deposited samples.

In addition to cell assignments, the web server allows users to view the metadata on which the assignment is based, view the ontology terms that are enriched for their data and the

distribution it predicts, and compare the expression of genes in the new profiled cells to genes identified as DE for the various cell types. The web server also clusters the cells and plots a 2D dimensionality reduction plot to compare the expression of the users' cells with all prior cell types it stores. Applying the method to analyze recent neurodegeneration data led to the identification of significant differences between cell distributions of healthy and diseased mouse models, with the largest observed difference being the set of immune-related cell types that are more prevalent in the diseased mouse. Our method also revealed additional upregulated immune-related genes in the late-stage neurodegenerative cells that were classified as "macro-phage" cells.

While the pipeline was able to process several of the datasets we identified on public repositories, not all of them could be analyzed. Specifically, many studies lacked raw scRNA-seq reads, and thus could not be processed via our uniform expression quantification pipeline. Though we were able to find author-processed expression data for many studies, usage of this data is complicated by different gene selections, data format differences (e.g. RPKM vs. FPKM vs. TPM vs. read counts), and more. Additionally, several studies profiled thousands of cells but published far fewer raw data files, with each raw data file containing reads from hundreds or thousands of cells but no metadata that allows each read to be assigned to a unique cell.

In addition to issues with processing data that has already been profiled and deposited, we observed that cell-type distribution in our database is still very skewed. While some cell types are very well represented ("bone marrow cell": 6283 cells, "dendritic cell": 4126 cells, "embryonic stem cell": 2963 cells) others are either completely missing or were only represented with very few samples ("leukocyte": 12 cells, "B cell": 22 cells, "microglial cell": 44 cells, "cardiac muscle cell": 72 cells). Such skewed distribution can cause challenges to our method leading to cells being assigned to similar, but not the correct, types. These are still the early days of scRNA-seq analysis with several public and private efforts to characterize cell types more comprehensively. Our dataset retrieval and processing pipeline (including cell-type assignments) is fully automated and we expect that once more

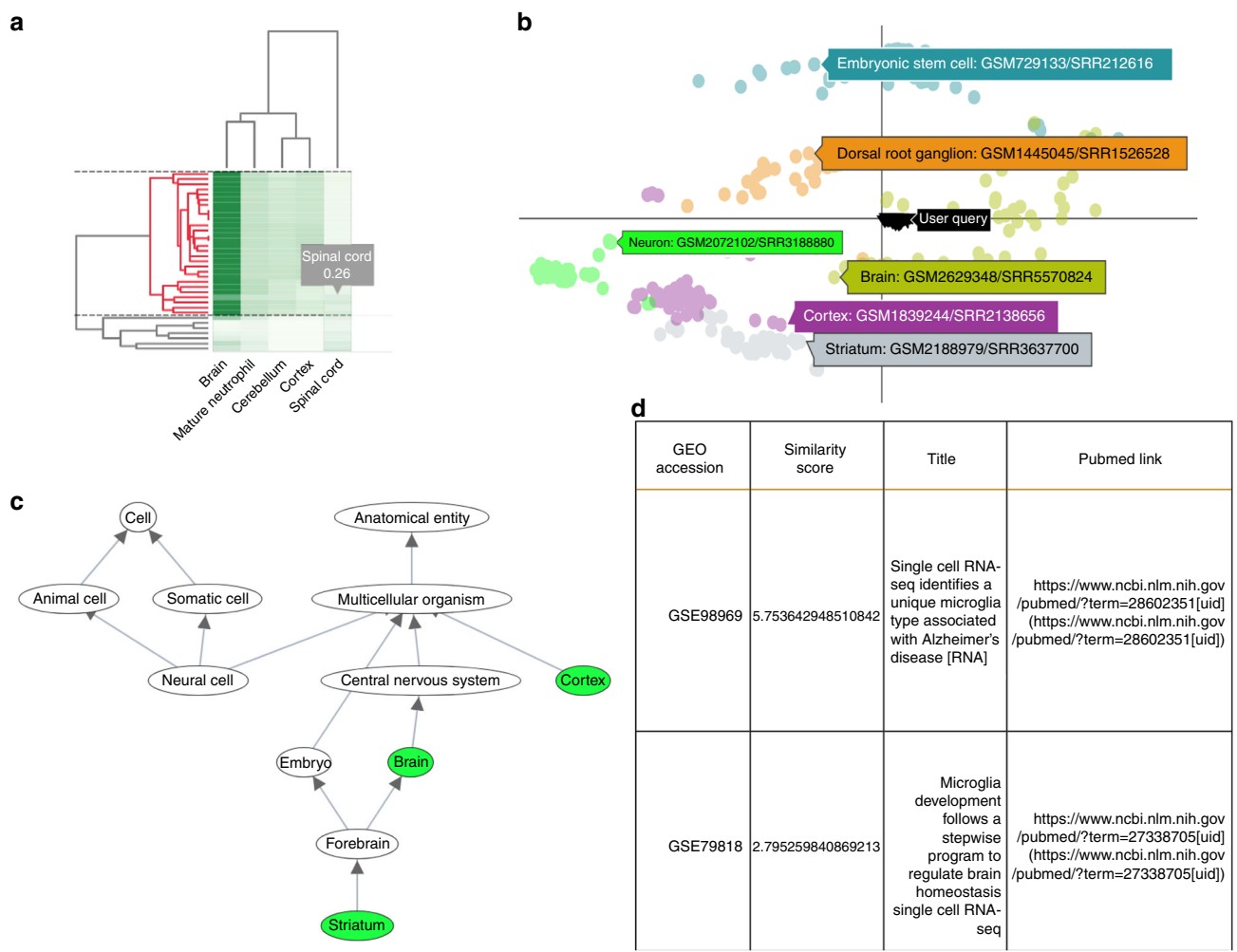

**Fig. 5** The scQuery web server. **a** Cluster heatmap of the nearest neighbor results for a query consisting of 40 "brain" and 10 "spinal cord" cells. The horizontal dashed lines demarcate the currently selected cluster and the corresponding dendrogram sub-cluster is highlighted in red. **b** 2D scatter plot of the selected sub-cluster (shown as inverted triangles and tagged as "User Query") along with a handful of other cell types whose tags show cell-type information and GEO submission ids for a single cell from each cluster. **c** Ontology DAG depicting the retrieved cell types in green while the nodes in gray visualize the path to the root nodes (which reflects paths of cellular differentiation as well as other biological relationships). **d** Metadata table for the retrieved hits displaying the GEO accession id, similarity score, publication titles, and their respective pubmed links

experiments are available they would be added to the database and server. We believe that as more data accumulates the accuracy of scQuery would increase making it the tool of choice for cell-type assignment and analysis.

## Methods

**Data collection and preprocessing**. We selected a mouse gene set of interest based on the NCBI Consensus CDS (CCDS) which contains 20,499 distinct genes. For genes with multiple isoforms, we consolidate all available coding regions (Supporting Methods). To search for scRNA-seq datasets, we queried the NCBI Gene Expression Omnibus (GEO) and the ArrayExpress database for mouse single-cell RNA-seq series (See Supporting Methods for the queries we used). We then download metadata for each series returned in this query and parse this metadata to identify the distinct samples that comprise each series. We examined the metadata for each sample (e.g., library strategy, library source, data processing) and exclude any samples that do not contain scRNA-seq data.

We next attempted to download each study's raw RNA-seq reads and for those studies for which these data are available, we developed a pipeline that uniformly processed scRNA-seq data. We use the reference mouse genome from the UCSC genome browser[17] (build mm10), and align RNA-seq reads with HISAT2[18] version 2.1.0. We align reads as single-end or paired-end as appropriate, and discard samples for which fewer than 40% of reads align to coding regions. We represent gene expression using RPKM. See Supplementary Fig. 3 for a histogram of read counts per series.

**Labeling using Cell Ontology terms**. We use the Cell Ontology (CL)[19] available from http://obofoundry.org/ontology/cl.html to identify the specific cell types which are represented in our GEO query results. We parsed the ontology terms into a directed acyclic graph structure, adding edges between terms for "is_a" and "part_of" relationships. Note that this choice of edge direction means that all edges point toward the root nodes in the ontology.

We use the name and any available synonyms for each ontology term to automatically identify the matching terms for each sample of interest (Supporting Methods). This produces a set of ontology term hits for each sample. We filter these ontology term hits by excluding any terms that are descendants of any other selected terms (e.g., term CL:0000000 "cell" matches many studies), producing a set of "specific" ontology terms for each sample—for any two nodes $u$ and $v$ in such a set, neither $u$ nor $v$ is a descendant of the other in the ontology.

**Dimensionality reduction**. Most current analysis methods for scRNA-seq data use some form of dimensionality reduction to visualize and analyze the data, most notably PCA and similar methods[20,21] and t-SNE[22]. Past work has shown that while such methods are useful, supervised methods for dimensionality reduction may improve the ability to accurately represent different cell types[10].

Using neural networks for dimensionality reduction has been shown to work well as a supervised technique to learn compact, discriminative representations of data[23]. The original, unreduced dimensions form the input layer to a neural network, where each dimension is an input unit. After training the model towards a particular objective (such as classification), the last hidden layer, which is typically much smaller in the number of units than the input layer, may be taken as a

reduced dimensionality representation of the data. These learned features are referred to as neural embeddings in the literature, and here we tested a number of different neural network architectures which either explicitly optimize these neural embeddings (for example, siamese[24] and triplet networks[12]) or those that only optimize the label accuracy. All neural networks we used were implemented in Python using the Keras API[25].

**Neural network architectures**. Prior work showed that sparsely connected NN architectures based on protein interaction data can be more effective in determining cell types when compared to dense networks[10]. Here we further studied other NN networks architectures and compared their performance to the PPI and dense networks. First, we looked at another method to group genes based on the Gene Ontology (GO)[26]. To construct a hierarchical neural network architecture that mirrors the structure of GO, we associate input genes with GO nodes. Multiple genes are associated (and connected to) the same node. We use this grouping of the input genes as the first hidden layer of a neural network. Nodes in the next hidden layer will be constructed from GO nodes that are descendants of nodes in the prior layer. We continue this process until the last hidden layer has the desired number of nodes (the size of our reduced dimension). The final result is the network depicted in Supplementary Fig. 13. See also Supporting Methods.

**Siamese architectures trained with contrastive loss**. The NNs discussed above indirectly optimize the neural embedding layer by optimizing a classification target function (correct assignment of scRNA-seq data to cell types). A number of NN architectures have been proposed to explicitly optimize the embedding itself. For example, siamese neural networks[11,24] (Supplementary Fig. 10) consist of two identical twin subnetworks which share the same weights. The outputs of both subnetworks are connected to a conjoined layer (sometimes referred to as the distance layer) which directly calculates a distance between the embeddings in the last layers of the twin networks. The input to a siamese network is a pair of data points and the output which is optimized is whether they are similar (same cell type) or not. The loss is computed on the output of the distance layer, and heavily penalizes large distances between items from the same class, while at the same time penalizing small distances between items from different class. Specifically, the network optimizes the following loss function:

$$\text{Contrastive loss} = \sum_{i=1}^{P} \left(Y^i\right) L_s\left(D^i\right) + \left(1 - Y^i\right) L_d\left(D^i\right) \tag{1}$$

Where :

$P$ is the set of all training examples (pairs of data points)

$Y$ is the corresponding label for each pair (1 indicates that

the pair belong to the same class, 0 indicates that each

sample in the pair come from different classes)

$D$ is the Euclidean distance between the points in the pair

computed by the network

$$L_s(D) = \frac{1}{2}(D)^2 \tag{2}$$

$$L_d(D) = \frac{1}{2}(\max\{0, m - D\})^2 \tag{3}$$

Where $m$ is a margin hyperparameter, usually set to 1

Following the same motivations as siamese networks, triplet networks also seek to learn an optimal embedding but do so by looking at three samples at a time instead of just two as in a siamese network. The triplet loss used by Schroff et al.[12] considers a point (anchor), a second point of the same class as the anchor (positive), and a third point of a different class (negative). See Supporting Methods for details.

**Training and testing of neural embedding models**. We conduct supervised training of our neural embedding models using stochastic gradient descent. Although our processed dataset contains many cells, each with a set of labels, we train on a subset of "high confidence" cells to account for any label noise that may have occurred in our automatic term matching process. This is done by only keeping terms that have at least 75 cells mapping to them, and then only keeping cells with a single mapping term. This led to a training set of 21,704 cells from the data we processed ourselves (36,473 cells when combined with author-processed data). We experimented with tanh, sigmoid, and ReLU activations, and found that tanh performed the best. ReLU activation is useful for helping deeper networks converge by preventing the vanishing gradient problem, but here our networks only have a few hidden layers, so the advantage of ReLU is less clear. We also experimented with different learning rates, momentums, and input normalizations (see web server for full results).

Since our goal is to optimize a discriminative embedding, we test the quality of our neural embeddings with retrieval testing, which is similar to the task of cell-type inference. In retrieval testing, we query a cell (represented by the neural embedding of its gene expression vector) against a large database of other cells (which are also represented by their embeddings) to find the query's nearest neighbors in the database.

Accounting for batch effects is a central issue in studies which integrate data from many different studies and experimental labs[27,28]. Here, we adopt a careful training and evaluation strategy in order to account for batch effects. We separate the studies for each cell type when training and testing so that the test set is completely independent of the training set. We find all cell types which come from more than one study, and hold out a complete study for each such cell type to be a part of the test set (sometimes referred to as the "query" set in the context of information retrieval). Cell types that do not exist in more than one study are all kept in the training set. For our integrated dataset, our training set contained 45 cell types, while our query set was a subset of 26 of the training cell types. After training the model using the training set, the training set can then be used as the database in retrieval testing.

**Evaluation of classification and embeddings**. In both training and evaluation of our neural embedding models, we are constantly faced with the question of how similar two cell types are. A rigid (binary) distinction between cell types is not appropriate since "neuron", "hippocampus", and "brain" are all related cell types, and a model that groups these cell types together should not be penalized as much as a model that groups completely unrelated cell types together. We have thus extended the NN learning and evaluation methods to incorporate cell type similarity when learning and testing the models. See Supporting Methods for details on how these are used and how they are obtained.

**Differential expression for cell types**. We use the automated scRNA-seq annotations we recovered to identify a set of differentially expressed genes for each cell type. Unlike prior methods that often compare two specific scRNA-seq datasets, or use data from a single lab, our integrated approach allows for a much more powerful analysis. Specifically, we can both focus on genes that are present in multiple datasets (and so do not represent specific data generation biases) and those that are unique in the context of the ontology graph (i.e., for two brain related types, find genes that distinguish them rather than just distinguishing brain vs. all others).

Our strategy, presented in Supplementary Algorithm 1, is DE-method agnostic, meaning that we can utilize any of the various DE tools that exist. In practice we have used Single-Cell Differential Expression (SCDE) here[29]. For this, we used read counts rather than RPKM, as SCDE requires count data as input. This method builds an error model for each cell in the data, where the model is a mixture between a negative binomial and a Poisson (for dropout events) distribution, and then uses these error models to identify differentially expressed genes. We also tried another method, limma with the voom transformation[30], but the list of DE genes returned was too long for meaningful analysis (many of the reported DE genes had the same $p$-value). Results of our comparison of SCDE and limma-voom are shown in Supplementary Table 4.

Another key aspect of our strategy is the use of meta-analysis of multiple DE experiments. The algorithm attempts to make the best use of the integrated dataset by doing a separate DE experiment for each study that contains cells of a particular cell type, and then combines these results into a final list of DE genes for the cell type. See Supporting Methods for the details of this meta-analysis.

**Large-scale query and retrieval**. To enable users to compare new scRNA-seq data to the public data we have processed, and to determine the composition of cell types in such samples, we developed a web application. Users download a software package available on the website to process SRA/FASTQ files. The software implements a pipeline that generates RPKM values for the list of genes used in our database and can work on a PC or a server (Supporting Methods).

Once the user processes their data, the data are uploaded to the server and compared to all studies stored in the database. For this, we first use the NN to reduce the dimensions of each of the input profiles and then use approximate nearest neighbor approaches to match these to the data we have pre-processed as we discuss below.

Since the number of unique scRNA-seq expression vectors we store is large, an exact solution obtained by a linear scan of the dataset for the nearest neighbor cell types would be too slow. To enable efficient searches, we benchmarked three approximate nearest neighbor libraries: NMSLib[31], ANNoY (https://github.com/spotify/annoy), and FALCONN[32]. Benchmarking revealed that NMSLib was the fastest method (Supplementary Table 3). NMSLib supports optimized implementations for cosine similarity and L2-distance based nearest neighbor retrieval. The indexing involves creation of hierarchical layers of proximity graphs. Hyperparameters for index building and query runtime were tuned to trade-off a high accuracy with reduced retrieval time. For NMSLib, these were: $M = 10$, efConstruction $= 500$, efSearch $= 100$, space $=$ "cosinesimil", method $=$ "hnsw", data_type $=$ nmslib.DataType.DENSE_VECTOR, dtype $=$ nmslib.DistType. FLOAT. Time taken to create the index: 2.6830639410000003 secs. Hyperparameters tuned for the ANNOY library were: number of trees $= 50$,

search_k_var = 3000. Time taken to create the index: 1.3495307050000065 secs. For FALCONN, a routine to compute and set the hyperparameters at optimal values was used. This calibrates K (number of hash functions) and last_cp_dimensions. Time taken to create the index: 0.12065599400011706 s.

**Visualizing query results**. We use the approximate nearest neighbors results to compute a similarity measure of each query cell to each ontology term. This is done by identifying the 100 nearest neighbors for each cell and determining the fraction of these matches that belong to a specific cell type. This generates a matrix of similarity measure entries for all query cells against all cell types which is presented as a hierarchical clustering heatmap (Fig. 5a). All visualizations are based on this matrix.

For each query cell $q_i$ and nearest neighbor $n_k$, we calculate the similarity score as:

$$s_{n_k}^{q_i} = 1/(1 + D(q_i, n_k))$$

Where $k \in [1, 100]$ and $D$ is the euclidean distance function.

We sum (over the nearest neighbors) the similarities to a specific cell type $ct$ to obtain a cumulative similarity score of the query to that cell-type:

$$S_{ct}^{q_i} = \sum_k \left( s_{n_k}^{q_i} * 1_{ct}(n_k) \right)$$

Where

$$1_{ct}(n_k) = \begin{cases} 1 & \text{if } n_k \text{ is cell type } ct \\ 0 & \text{otherwise} \end{cases}$$

Thus, we obtain for each query a vector of similarity scores against all cell types. Finally, we normalize the vector such that the cell-type-specific cumulative similarities sum to 1. Each normalized vector forms a row in the hierarchical heatmap.

We also perform further dimensionality reduction of the query via PCA to obtain a 2D nearest-neighbor style visualization against all cell types in the database and generate the ontology subgraph that matches the input cells. Users can click on any of the nodes in that graph to view the cell type associated with it, DE genes related to this cell type, and their expression in the query cells.

In addition to matching cells based on the NN reduced values, we also provide users with the list of experiments in our database that contain cells that are most similar to a subset of uploaded cells the user selects. This provides another layer of analysis beyond the automated (though limited) ontology matching that is based on the cell types extracted for the nearest neighbors.

Finally, users can obtain summary information about cell-type distribution in their uploaded cells and can find the set of cells matched to any of the cell types in our database.

**Code availability**. Code for our preprocessing (alignment/quantification) pipeline is available at https://github.com/mruffalo/sc-rna-seq-pipeline. Code for training and evaluation of our neural network models are available at https://github.com/AmirAlavi/scrna_nn. Code for our differential expression analysis to find cell-type-specific genes is available at https://github.com/AmirAlavi/single_cell_deg.

## Data availability

The reprocessed data that support the findings of this study are publicly available online at https://scquery.cs.cmu.edu/processed_data/.

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

## Acknowledgements

Work partially supported by NIH Grant 1R01GM122096, the Pennsylvania Department of Health (Health ResearchNonformula Grant CURE Award 2015) and a James S. McDonnell Foundation Scholars Award in Studying Complex Systems to Z.B.-J and by NIH grant 1F32CA216937 to M.M.R. This work used the Extreme Science and Engineering Discovery Environment (XSEDE), which is supported by National Science Foundation grant number ACI-1548562. We thank Hongyu Zheng (Computational Biology Dept., CMU) for his work on cell-type similarity calculation. We also thank Andreas Pfenning for his suggestions and advice on the mouse brain case study analysis.

## Author contributions

Conceptualization: Z.B.-J; Methodology: Z.B.-J., A.A., and M.R.; Software: A.A., M.R., A.P. and Z.H.; Formal analysis: A.A.; Investigation: Z.B.-J., A.A., and M.R.; Data curation: M.R. and Z.H.; Writing—original draft: Z.B.-J., A.A., M.R., A.P. and Z.H.; Writing—review and

editing: Z.B.-J., A.A., M.R., A.P. and Z.H.; Visualization: A.A., M.R., and A.P.; Supervision: Z.B.-J., A.A. and M.R.; Project administration: Z.B.-J.; Funding acquisition: Z.B.-J, M.R.

## Additional information

**Competing interests:** The authors declare no competing interests.

