## [Peer Review file · Nature Communications]

Reviewers' comments:

Reviewer #1 (Remarks to the Author):

The study describes some new ideas for single cell analysis, but the manuscript and web-site appear incomplete.

Major concerns

It would be great if all reprocessed datasets are made available for download and exploration on the web-site having separate pages for each study. It is expected that more features should be available to explore the processed public datasets.

Not clear how the "text analysis" was used to assign cell type to single cells.

Overall, the web-site is not very functional and useful yet.

Submitted a list of genes to the server but received zero overlap with different ranks of returned terms. This is misleading and should be fixed. There should be instructions about the type of gene ID the application expects. Is there support for multiple organisms?

The example for brain on the website should also provide the data in text format so users can see the required format.

Example should be added to the gene list submission form as well.

It would be great to provide cloud alignment support, so users can have their data processed by the pipeline by the authors.

The alignment documentation on GitHub is lacking detail and clarity. Examples should be added. How do you specify the organism and other parameters for the aligner?

Ontology graph did not display anything with the loaded example.

The hierarchical clustering heatmap on the site is difficult to interpret.

Can you please show that read counts work better than RPKM rather than referring to studies from 2010?

It is unclear how DE genes were determined. This is important to explain clearly upfront. The SCDE method needs to be explained, acronym spelled out, and comparison to other methods is needed.

>7000 DE genes sounds too large and not specific. Consider adjusting the number of DE genes dynamically.

The GSEA method is known to perform poorly. It should be benchmarked with other methods.

It is unclear how the Gene Ontology was used. What branch? Which terms?

Discussing the querying tool in the middle of the article appears odd. The case study should be moved up and the tool description to the end.

The identification of more immune cells in the case study should be highlighted.

What is Algorithm ??

Minor suggestions

The introduction has only two lean paragraphs about prior work while the rest of the introduction is used to describe the application and should move to the results section.

The plots on the website use mixed font and are not well formatted nor responsive.

Genelist should not be one word on the site.

Can't read text in figures 1A 1E and other figures.

Figure 3 is a table. What about the other terms? Can you show all term in this figure/table?

Reviewer #2 (Remarks to the Author):

Referee report for: "scQuery: a web server for comparative analysis of single-cell RNA-seq data" by Alavi et al.

The authors present a database of single-cell RNAseq measurements for public use and query. For this purpose they developed an automated pipeline to collect raw data from public data repositories, and run all data through an analysis procedure they have developed, using a supervised neural network for compact data representation. They then construct the database using ~500 studies representing a total of ~300 cell-types and >60,000 cells. The authors demonstrate the use of this tool by querying new data from a study about microglial cells in an Alzheimer disease model.

The study represents a brave effort to collect and analyze scRNAseq data on a global scale with standard and systematic pipeline – in general a good idea that would be of value to the community. Because of many technical aspects, the task is however a non-trivial problem.

Some comments that I think can improve the manuscript:

1. It is not clear to me what is the aim of the study; who is the target user? As an expert in scRNAseq analysis I would not use such a tool because such generic analysis may lose specific details relevant for the specific experiment. To me it seems that this type of tool can be relevant as exploratory analysis for the non-expert user. Moreover, if the aim is just to compare samples based on similarity by KNN, this might be overkill. Maybe the authors can present their vision of what one can do with so much data – that would not be possible otherwise. What can autoencoders teach us? What is the interpretation of the neural embedding?

2. My biggest technical issue is that the authors completely ignore the main type of scRNAseq data produced today by the community, which is based on unique molecular identifier (UMI) methods. Instead, they use raw reads data (or RPKM), which limits the type of data to be "smartseq"-like (data that includes gene body, and does not have UMI). If the authors could handle also UMI data, the size of the database would grow to millions of cells and potentially become very powerful.

3. It is not clear how batch effects or other technical artifacts may affect the results. Recently, a number of new studies about merging datasets were published (Butler et al Nat. Biotech. 2018, Hahverdi et al Nat. Biotech. 2018 for example), but this central issue is not currently discussed in the manuscript.

Minor comments:

1. Gene names should be italic.
2. The graphical presentation of the figures can be dramatically improved. For example, Figure 2 is not very exciting and font size of the x-axis is very difficult to read, can go to Supplementary or turn into a panel in another figure. Figure 3 is actually a table, the terms and values are very difficult to follow and the overall conclusion is not clear. Many of the fonts in figure 5-6 are not readable, and in general the use of "website view" is not helpful, figures are better made dedicated for journal publication.
3. The authors say they analyzed ~500 datasets. The merged data has 60,000 cells, meaning the average study has just 120 cells. This seems like a very small number for published data.

Reviewer #3 (Remarks to the Author):

The manuscript discusses a very important topic of reusing the existing scRNAseq data when getting the new data to compare them with each other and to probably assign cells of the new experiments to the cell types from the old experiment. Overall, the manuscript is well written and a lot of work has been done. However, there are few concerns that I would the authors to address:

- How scQuery can deal with the rare cell-types? When the rear cell types can be in either the new or the old datasets.
- Is there a way in scQuery of marking cells as "Unassigned cells"? If yes, how does it work?
- With new 10X experiments the total amount of cells in a single experiment can be larger than the total number of cells in scQuery. Have you thought about how to deal with the 10X data since it's becoming very popular and standard way of doing scRNAseq in addition to Smartse2?
- In Fig.3: could you please explain the low numbers in the stem cell and embryo categories?
- In Fig.3: it does not look like PCA and original data are significantly outperformed by the ANN. Could you please comment on this?
- In the web app for the example dataset not all of the results tabs show the actual results (e.g. Cell Types, Low dimensional mapping etc).
- In the web app the overall user interface can be significantly improved. At the moment it's not clear at all how the web site works and what it does for the end user.

Reviewer 1:

Major:

It would be great if all reprocessed datasets are made available for download and exploration on the web-site having separate pages for each study. It is expected that more features should be available to explore the processed public datasets.

We have added additional metadata to the processed expression data that is available for download. We have added an “index” table on the website, listing each distinct study which is included in our database, the number of usable cells from that study, the starting and ending row indexes for that study, and a link to the GEO or ArrayExpress pages for that study (https://scquery.cs.cmu.edu/processed_data/index_table/). We have also uploaded an additional version of the processed expression data with hierarchical row labels, which allows for efficiently querying/downloading data for a specific study from our database without obtaining the entire multi-gigabyte HDF5 file (https://scquery.cs.cmu.edu/processed_data/). **Not clear how the "text analysis" was used to assign cell type to single cells.**

As the reviewer suggested we have added a section to the supplement to explain how we extracted cell type information from the GEO datasets. Please see the section titled “Labeling of single cell experiments using Cell Ontology terms” for details on the algorithm used to assign cell ontology terms to GEO and ArrayExpress experiments.

Overall, the web-site is not very functional and useful yet.

We have made several changes to the website following the comments from the reviewers. Briefly:

- Added an example input for the DE analysis
- Change the way tabs are presented to only enable those that can be used given current user selection
- Added more “help” information for specific tabs
- Added a short video, embedded on the web server front page, in which we show how to use the various website functionalities

Submitted a list of genes to the server but received zero overlap with different ranks of returned terms. This is misleading and should be fixed. There should be instructions about the type of gene ID the application expects. Is there support for multiple organisms?

Currently all models and analysis only support *M. musculus*. As the reviewer suggested we have made several changes to the DE list input option:

- We now validate the user's input gene list: if we encounter an invalid identifier, we notify the user.
- We added radio buttons for gene ID types, and currently support gene symbols and entrez IDs.
- The results are now ranked correctly.
- The placeholder text now reads: "Paste tab-delimited differentially expressed mouse genes, select type of gene ID and submit".
- We have added a sample input button for DE genes

The example for brain on the website should also provide the data in text format so users can see the required format.

As we discuss in the paper and state on the website, the uploaded files must be outputs of our processing pipeline. As we now explain in the supplement, we have made this choice since alignment and initial processing times dominate computational resources and if we provide this as a service a large experiment can block the use of the website by all others. Users can download and use the pipeline on their own machines (we tested it successfully on several machines). Given this, the uploaded files are actually binary (to significantly reduce storage space) and so showing users the format of these files will not help (as opposed to the gene list that we agree requires an example input that we have now added as discussed above). This is now clarified in the manual and mentioned in the video tutorial.

Example should be added to the gene list submission form as well.

As the reviewer suggested we have now added a one-click example gene list submission button. Clicking the button will populate the input box with an example list of genes. The user can then submit this list, or modify/inspect it before doing so.

It would be great to provide cloud alignment support, so users can have their data processed by the pipeline by the authors.

We agree that an end-to-end solution would be ideal for users. However, the initial processing is the most computationally intensive aspect of the analysis. While most users can be supported, if a single user uploads a large dataset (tens of thousands of cells) it can create a bottleneck that will limit access to the web server for a long time. Since such large scale users often have access to their own computing resources we believe that a standalone pipeline provides a good balance between the need to standardize input, the needs of the less experienced user (who usually would only need to process a few to hundreds of cells which can be done on a personal computer), and large users with access to extensive computational resources. We are able to provide scQuery as a free service mostly because the querying of our large database can be efficiently done via our reduced dimension representation and fast approximate nearest neighbor techniques and we would like to keep it as lean and fast as possible.

The alignment documentation on GitHub is lacking detail and clarity. Examples should be added. How do you specify the organism and other parameters for the aligner?

As suggested we have significantly expanded the alignment documentation on GitHub, including descriptions of common workflows with multiple inputs: single- and paired-end FASTQ files, SRA files, and SRR IDs deposited in public repositories such as GEO. We have additionally expanded the alignment documentation to include information on specifying options to the HISAT2 aligner, with links to prebuilt indexes for various organisms and the specific index used to build our database of expression data.

Ontology graph did not display anything with the loaded example.

We believe this was because the navigation and use of the website was not clear: a user must click on a cluster in the initial heatmap to see results in other tabs. We have now improved the

UI of the web app to make this more apparent and added a video tutorial. Please see our response to the final reviewer comment at the end of this letter for more details.

The hierarchical clustering heatmap on the site is difficult to interpret.

We have added discussion of the clustering heatmap to the Methods under the heading “Visualizing query results”. We have also enhanced the website which now includes help information when “hovering” over the clustering results. We have also disabled specific tabs that are related to the clustering map to reflect the need to select a specific cluster for further analysis.

Can you please show that read counts work better than RPKM rather than referring to studies from 2010?

We agree that we did not provide a strong explanation for the use of read counts in the DE analysis and have now revised this section. Specifically, we tried a number of DE analysis algorithms and eventually selected SCDE which was designed to work with read counts and not RPKM data. Given the large number of cells (expression profiles), several of the DE methods assigned max significance values to a very large number of genes for each cell type which is of course, uninformative. SCDE was able to better rank genes and so we have used it for scQuery. We now clarify this in the text and rather than mentioning read counts, refer to the actual SCDE paper itself. We also added results from another tool we tried, limma-voom, to the supplements as we discuss below.

It is unclear how DE genes were determined. This is important to explain clearly upfront. The SCDE method needs to be explained, acronym spelled out, and comparison to other methods is needed.

As the reviewer suggested, we have added the following to the paper to explain how our DE method works. We emphasize here that Single Cell Differential Expression (SCDE) (Karchenko et al., 2014, PMID: 24836921) is not our method. We use SCDE within our algorithm for finding cell-type specific DE. In Algorithm 1 (which outlines how we determine DE genes), we have added a summary to the caption to help understand the pseudocode presented. In the Supporting Methods, we revised the sections titled “Preprocessing strategies for differential expression analysis” and “Meta-analysis of differential expression experiments” to better explain our DE analysis procedure in more detail.

We have also added text discussing our comparison of this method with other DE RNA-Seq methods (limma-voom). We briefly mention in the paper that for our case, the results from limma-voom are much less informative when compared to the results from SCDE. The following table (added as Supplementary Table S6) compares the most significant GO categories for the top 50 most significantly upregulated genes identified by limma-voom and SCDE for “retina”:

Cell Type	retina		Cell Type	retina	
Term ID	UBERON:0000966		Term ID	UBERON:0000966	
# of experiments	2		# of experiments	2	
GO ID	Name	p-value	GO ID	Name	p-value
GO:0050877	nervous system process	7.63e-09	GO:0045721	negative regulation of gluconeogenesis	3.03e-02
GO:0050953	sensory perception of light stimulus	7.63e-09	GO:0032379	positive regulation of intracellular lipid tr...	3.03e-02
GO:0007601	visual perception	7.63e-09	GO:0032385	positive regulation of intracellular choleste...	3.03e-02
GO:0007423	sensory organ development	8.69e-09	GO:0061179	negative regulation of insulin secretion invo...	3.03e-02
GO:0060041	retina development in camera-type eye	8.69e-09	GO:1905600	regulation of receptor-mediated endocytosis i...	3.03e-02
GO:0007600	sensory perception	4.09e-08	GO:0010888	negative regulation of lipid storage	3.03e-02
GO:0001654	eye development	3.96e-07	GO:0032382	positive regulation of intracellular sterol t...	3.03e-02
GO:0003008	system process	6.67e-07	GO:1905602	positive regulation of receptor-mediated endo...	3.03e-02
GO:0009584	detection of visible light	6.86e-07	GO:0045475	locomotor rhythm	3.03e-02
GO:0043010	camera-type eye development	1.23e-06	GO:0090118	receptor-mediated endocytosis involved in cho...	3.34e-02

(a) SCDE

(b) limma-voom

As can be seen, enrichment results from limma-voom (b) are much less specific when compared to the same enrichment analysis done with SCDE (a).

>7000 DE genes sounds too large and not specific. Consider adjusting the number of DE genes dynamically.

We agree that a large number of DE genes may be less beneficial than a small one. We note that the definition of DE is somewhat arbitrary and depends on a cutoff. Given the size of the data we analyze (thousands of expression profiles for each cell compared to relatively small number in bulk expression analysis) it may be expected that some cells would look different from several other cells leading a large number of genes with an adjusted p-value < 0.05. Still, we agree with the reviewer that such a large list may not be what users are looking for. Thus, in scQuery we do not use all DE genes but just the top 100 up and down regulated genes for each cell type which is much more manageable. If you click on the DE genes tab in scQuery you would only see the expression for this top 100. We now emphasize this in Methods. Note that we still keep the p-values for all DE genes and so can easily rank them to apply other cutoffs and reduce the list as well. GO analysis results presented in the paper for the DE genes are based on the top 50, which, as we show, usually represent the correct function associated with each of the cells.

The GSEA method is known to perform poorly. It should be benchmarked with other methods.

We believe we did not explain the actual way we used GSEA. Rather than relying on their curated set, we simply used their method to determine enrichment p-values for GO gene sets (the main difference is that such methods take ordering into account). Our actual analysis was based on “GO biological process” assignments for genes and while we agree that these are not perfect, and certainly miss a lot of information, they are still one of the most popular methods to validate groupings of genes and DE analysis (Knott et al., *Nature* 2018, PMID: 29414946) (Martyn et al., *Nature Genetics* 2018, PMID: 29610478). In the revised version we have replaced all places in which we mention “GSEA” with “GO enrichment analysis”. We have also added a section in Supporting Methods, “Gene Ontology enrichment analysis”, to explain the details of the above GO enrichment analysis.

It is unclear how the Gene Ontology was used. What branch? Which terms?

To address this comment we added a new section in Supporting Methods (“Gene Ontology enrichment analysis”) which includes our parameters for this analysis. Specifically, we mention that we use the “Biological Process” branch of the ontology, and include all terms that have annotations for *M. musculus*.

Discussing the querying tool in the middle of the article appears odd. The case study should be moved up and the tool description to the end.

As suggested, we changed the order to move discussion of the web server to the end of Results and slightly changed the text of the case study based on this.

The identification of more immune cells in the case study should be highlighted.

As suggested we have added text to this effect to the figure caption and Results.

What is Algorithm ??

The missing Algorithm number has been fixed. Now it correctly points to “Algorithm 1” (which is in the Supporting Figures).

Minor:

The introduction has only two lean paragraphs about prior work while the rest of the introduction is used to describe the application and should move to the results section.

As suggested we greatly shortened the Introduction part that discusses the details of the methods we developed and our pipeline. This is now explained at the first section of Results and in Figure 1.

The plots on the website use mixed font and are not well formatted nor responsive.

We have reformatted all of the plots on the website to use the same fonts as the rest of the site. In addition, we made the plots on the front page responsive to screen size. We also improved the usability of all scatter plots on the web server (now the user can show/hide specific cell types by clicking on the legend).

Genelist should not be one word on the site.

This was a typographical error. It has now been changed to “gene list”.

Can't read text in figures 1A 1E and other figures.

Figure 1A and 1E have both been enhanced and enlarged so that the fonts are readable. We have also enhanced and enlarged Figure 5 (now Figure 4) and enhanced Figure 6 (which is now split and renamed into Figure 3 and Figure S6). We have also enlarged the fonts in Figure S4 and S6 (now S5).

Figure 3 is a table. What about the other terms? Can you show all term in this figure/table?

We have relabeled this Figure as “Table 1”. There are 26 terms in the retrieval testing that we conducted, so we are unable to show them in the text due to space. However, we now include the full results with all of the terms in an excel file on the website at this address:

<https://scquery.cs.cmu.edu/publications/>.

Reviewer 2:

Major:

It is not clear to me what is the aim of the study; who is the target user? As an expert in scRNAseq analysis I would not use such a tool because such generic analysis may lose specific details relevant for the specific experiment. To me it seems that this type of tool can be relevant as exploratory analysis for the non-expert user. Moreover, if the aim is just to compare samples based on similarity by KNN, this might be overkill. Maybe the authors can present their vision of what one can do with so much data – that would not be possible otherwise.

The reviewer is correct that the main goal of scQuery is to serve experimentalists who are performing exploratory analysis of new scRNA-Seq studies or re-analyzing existing studies. In such cases one of the first questions that are asked are “what are the cell types that are present in the data I just profiled?” To date, answering such a question (even after dimensionality reduction and clustering) required manual analysis of marker genes for each cluster. In several cases it was challenging, or even impossible, to fully annotate all clusters (for example when markers were unknown or when the researcher was not an expert on other cell types). scQuery solves this problem by automating the annotation process based on prior deposited samples. This has several advantages. First, since it is automated, it is much faster, and often more accurate, for non-expert users. Second, it uses an unbiased approach and does not require the manual selection of specific markers. Third, it provides users with several follow up analyses options including the ability to determine the distribution of cell types, the set of DE genes for each of these cell types, grouping of the cells based on their predicted type etc. We have added this to the Discussion.

We also believe that a tool that can utilize hundreds of thousands of previously profiled cells is not a trivial issue or an overkill. Even the comparison itself is not trivial since, as we discuss, to make it efficient requires novel dimensionality reduction methods and sophisticated computational methods (including neural networks and approximate nearest neighbors).

What can autoencoders teach us? What is the interpretation of the neural embedding?

We agree with the reviewer that this is an interesting question. While the major focus of this work is on the web server (and so the NN serves as a means to an end, which is reducing the dimension of the data to improve retrieval) we still observe that in some cases architectures with interpretable internal nodes may provide an advantage. In past work (Lin et al., 2017, PMID: 28973464) we tested several ways to encode meaning in the networks including creating nodes based on TF-gene interactions and nodes that capture protein-protein interactions. Here we extended these by also testing GO based architectures. In all cases the interpretation is that genes that are involved in a common process (pathway, function, regulation) may be more informative as a group rather than individuals. For example, if we have a specific GO function node, that node can be used to correctly associate expression profiles with cell types even if some of the genes in the pathway are not expressed in that sample (due to noise or dropouts) since the node aggregates information from all genes in that pathway/function. While with enough data we can hope that the network would be able to learn this on its own (and in some cases it does, see Lin et al., 2017), by using prior biological knowledge we explicitly encode this in our models making them more interpretable. We have added a section to the supplement to discuss these biologically based architectures.

My biggest technical issue is that the authors completely ignore the main type of scRNAseq data produced today by the community, which is based on unique molecular identifier (UMI) methods. Instead, they use raw reads data (or RPKM), which limits the type of data to be “smartseq”-like (data that includes gene body, and does not have UMI). If the authors could handle also UMI data, the size of the database would grow to millions of cells and potentially become very powerful.

The reviewer is correct that most (though not all) data used by scQuery so far is based on our uniform processing of deposited reads data. We believe that by processing deposited data in a uniform manner we can overcome many problems related to different list of genes, different normalization and quantification methods etc. However, as the reviewer mentions not all data can be processed in the same way. While this may have not been clear in our initial submission, we do use non-uniformly processed data, including UMI data. As we show in Figure 2, 29,216 expression profiles used by scQuery are author processed (roughly 20% of our current data). Author processed data is used by attempting to normalize non raw expression values (i.e. expression values provided by the uploader) as discussed in the supplement (see section titled “Missing data imputation”). UMI data is, and would continue to be, incorporated into scQuery using this process. As for the numbers cited by the reviewer, we note that we cannot use several datasets for various reasons which are listed in the paper including inability to match cells to specific ontology terms, the fact that the data is not deposited in public repositories, the fact that not enough genes from the reference list were used by the authors and more. While this indeed limits the number of cells used by scQuery, it also means that the quality of the data we do use is high. We believe that as more and more datasets are generated the number of cells in scQuery would increase (indeed, just in the three months from submission to this current revision we have almost doubled the number of cell profiles used by scQuery). So while it may take time to get a million cells, it would not take too long.

It is not clear how batch effects or other technical artifacts may affect the results.

Recently, a number of new studies about merging datasets were published (Butler et al Nat. Biotech. 2018, Hahverdi et al Nat. Biotech. 2018 for example), but this central issue is not currently discussed in the manuscript.

We agree that it is extremely important to account for batch effects when integrating datasets from various studies. In our case, we take great care in the way we train and evaluate our models to account for this. For the model selection phase (after training), we evaluate each model using a retrieval test to simulate the real-world use of these models. For this, we use a held-out dataset (also known as a query set in the context of information retrieval): For each cell type that has been profiled by more than one study, we completely hold out one of its studies to be part of the query set. In this way, we prevent choosing a model that learns to place cells of the same type near each other in embedded space because they come from the same study. While we mentioned this process in the Methods under the heading “Training and testing of neural embedding models” (last paragraph), we agree with the reviewer that this is indeed a central issue, and must be highlighted. To this end, we added a sentence explaining that accounting for batch effects was the reason for adopting this training and evaluation strategy.

Minor:

Gene names should be italic.

All gene names have now been italicized.

The graphical presentation of the figures can be dramatically improved. For example, Figure 2 is not very exciting and font size of the x-axis is very difficult to read, can go to Supplementary or turn into a panel in another figure. Figure 3 is actually a table, the terms and values are very difficult to follow and the overall conclusion is not clear. Many of the fonts in figure 5-6 are not readable, and in general the use of “website view” is not helpful, figures are better made dedicated for journal publication.

As suggested, we revised Figure 2 to contain more information about the specific usable data over time, which we believe would make it more interesting and informative. We also revised the x axis fonts to make it more readable. We note that the new figure is also updated to reflect studies that were collected between the time of submission and the time of this revision. We have also renamed Figure 3 as Table 1. We have also added more text to the caption of Fig 3 (now Table 1) to make the overall conclusion of this table more clear. Fig 5c (now Fig 4c) has been replaced with a smaller tree so that nodes are now easily readable. Fig 5d has been removed to make space to enlarge the other panels. Fig 5e (now Fig 4d) has been enlarged so that its information can be read more easily. Fig 6c has been moved to a separate figure in the supplement (Figure S6), and has been revised to only show the genes we refer to in the text, rather than showing many genes in a crowded figure. This allowed us to make the fonts bigger. Finally, while we agree that paper figures should not necessarily be screenshots from tools, in this specific case our goal is to introduce the tool itself so we believe that a figure that included screen captures is essential for such a paper.

The authors say they analyzed ~500 datasets. The merged data has 60,000 cells, meaning the average study has just 120 cells. This seems like a very small number for published data.

Note that as we show, our data contains studies spanning almost 5 years. Initial datasets were indeed smaller than current ones. In addition, as we now mention in Results, many studies contain more cells than are actually usable in our analysis. We have a number of QA requirements for each cell (% of reads that could be mapped, ability to correctly associate cells with ontology terms) and in addition some of the short read data deposited lacks information required to assign reads to individual cells. In these cases we try to use data processed by authors, often in the form of matrices with genes and cells as rows/columns, but usage of this author-processed data has several disadvantages that we intended to avoid with our uniform processing pipeline. Notably, individual study authors’ selections of genes, gene locations, file

formats, expression data format (RPKM, FPKM, TPM, read counts) can all have a large effect on the number of cells we can use from each study.

Reviewer 3:

How scQuery can deal with the rare cell-types? When the rare cell types can be in either the new or the old datasets.

We agree with the reviewer that this is a very interesting question, and is something that we are very interested in addressing directly in future work. Currently, scQuery is able to retrieve some rare cell types with high sensitivity, but has low performance for some other rare cell types. This low performance is due to lack of training data for these rare cell types. Based on this comment we further investigated this issue. We have added a new table (Table S3 “Retrieval testing results for lowly represented cell types in our database”). This table provides a closer look at the performance on cell types which have less than 1% of cells in our training data. In this table, we show that even for such rare cell types, our neural embeddings are better than PCA and original data. We see very high performance for “fibroblast”, but we see poor performance for “embryo”. Overall, the average performance of all models is lower among the rare cell types in our database (comparing Table S3 to Table 1).

Is there a way in scQuery of marking cells as "Unassigned cells"? If yes, how does it work?

This is a very good point. While in general we do not flag cells as “unassigned”, users can indeed remove cells from the analysis. Whether some cells are unassigned or not depends on the threshold the user selects for matches (threshold bar is shown for many tabs in scQuery). All query cells are assigned a score for all cell types in the database. In some cases (where the query cell indeed matches a known database type) the score will be very high for one (or a few) cell types and low for all others. Cells that do not match any known type in the database (and so should be unassigned), are likely to be matched with low scores to several different cell types (scores sum to 1 and since their expression profile is likely different from all cell types we expect scores would be roughly distributed among many weak matches). Such cells would not be used in the different analyses performed (cell type distribution, DE analysis, ontology selection etc.). So implicitly the method can indeed deal with this issue. Explicitly dealing with it (that is, identifying novel cell types) is beyond the scope of this paper but we agree is an important and interesting question.

With new 10X experiments the total amount of cells in a single experiment can be larger than the total number of cells in scQuery. Have you thought about how to deal with the 10X data since it's becoming very popular and standard way of doing scRNAseq in addition to Smartseq2?

We agree that larger datasets would require longer time to process. However, the main time consuming aspect for us is the alignment of short reads rather than the specific number of cells. More recent (and larger in terms of cells) datasets are often using lower coverage per cell which reduces processing time. We have access to both a computer cluster and the Pittsburgh Supercomputing Center (PSC) and have used both for the processing. So far, we have not had trouble in our ability to process usable data, including some cells/series which contain over one billion short reads (the maximum read count we encountered for a study so far is 1,089,994,162). Our main challenge is often to find raw read data since, as we discuss in Methods, for some of the datasets we can only obtain processed data. Based on this comment we have added a new Figure (S3) which contains information about read counts in the data we processed.

In Fig.3: could you please explain the low numbers in the stem cell and embryo categories?

(In the following, we now refer to Fig 3 as Table 1)

The cell type for which we see the poorest retrieval performance in Table 1 is “embryo”. We believe that this is due to lack of training data. In Table 1, we can see that “embryo” has the fewest number of cells in our database (which are also used for model training). As mentioned above, we have added an additional Table S3 which shows retrieval performance for the rare cell types in our database, and “embryo” is indeed one such cell type with less than 1% of the total database population. We believe that as we accumulate more data, the performance for these rare cell types will improve. For “hematopoietic stem cell”, we see that some neural embedding models perform quite well, and others perform poorly. We call attention to this in the paper by mentioning that some models perform better on particular cell types than others, and that it is worth trying a few neural embedding models. However, we can see in Table 1 that in each cell type (each column), the best performing model (bolded number), is indeed one of our neural embedding models, not PCA or original data.

In Fig.3: it does not look like PCA and original data are significantly outperformed by the ANN. Could you please comment on this?

In Figure 3 (Now Table 1), we try to highlight the fact that PCA and original data are outperformed by the ANN. In the table, each column is the mean average precision scores for queries of a certain cell type. In each of these columns, the highest performing model is always one of our ANNs, never PCA or original data (the bolded number in each column is the best score). Finally, in the last column, we present a weighted average (weighted by number of queries of each cell type) of the scores across all cell types with more 1000 samples in the retrieval database. In this weighted average column, we can observe the dramatic difference between our best ANN (PT dense 1136 100, score 0.623) and PCA 100 (score 0.494) or original data (score 0.109). To determine the significance of this difference we have now added the results of a paired t-test that shows that our best model is significantly better than PCA 100 on a dataset for cell types with more than 1000 samples. These results are in the caption and the text. For this comparison we obtain a p-value of 1.253×10^{-41} .

In the web app for the example dataset not all of the results tabs show the actual results (e.g. Cell Types, Low dimensional mapping etc).

To see these the user has to select a sub-cluster from the hierarchical clustering tree (or select the entire tree) to further analyze. We apologize that the functionality of the website was not clearly explained. We have implemented many UI enhancements for the revision to help guide the user so that they will actually see results in the tabs for the example query as we discuss below.

In the web app the overall user interface can be significantly improved. At the moment it's not clear at all how the web site works and what it does for the end user.

We have made several changes to the website following the comments from the reviewers.

Briefly:

- Added an additional example input for the DE analysis
- Change the way tabs are presented to only enable those that can be used given current user selection
- Added more ‘help’ information for specific tabs
- Created a short video screencast that walks a user through using the website with an example query, and have embedded this video on the homepage (link to video on YouTube: <https://www.youtube.com/watch?v=s8D-YeXLAG0>).

REVIEWERS' COMMENTS:

Reviewer #1 (Remarks to the Author):

Reviewer #2 (Remarks to the Author):

The authors carefully respond to my comments and I think that overall the quality of the paper is improved. Such web resources are of important to the community of non-expert users in order to explore their datasets without the need for bioinformatics experts. As future work I hope that the authors will make the effort to make use of UMI datasets which are going to be a major player in the field of single-cell transcriptomics. I recommend to accept the paper for publication in Nature Communications.

Reviewer #3 (Remarks to the Author):

The authors have addressed all of my concerns. I recommend approval.